# Extrusion-Based Additive Manufacturing-Driven Design and Testing of the Snapping Interlocking Metasurface Mechanism *ShroomLock*

Philip Gloyer [†], Lucca Nikita Schek [†], Hans Lennart Flöttmann, Paul Wüst and Christina Völlmecke *

Stability and Failure of Functionally Optimized Structures, Institute of Mechanics, Technische Universität Berlin, Einsteinufer 5, 10587 Berlin, Germany; philip.gloyer@tu-berlin.de (P.G.); l.schek@tu-berlin.de (L.N.S.); h.floettmann@campus.tu-berlin.de (H.L.F.); paul.l.wuest@campus.tu-berlin.de (P.W.)

* Correspondence: christina.voellmecke@tu-berlin.de
† These authors contributed equally to this work.

**Abstract:** This study presents the manufacturing process-driven development of an interlocking metasurface; (ILM) mechanism for fused filament fabrication; (FFF) with a focus on open-source accessibility. The presented ILM is designed to enable strong contact between two planar surfaces. The mechanism consists of spring elements and locking pins which snap together when forced into contact. The mechanism is designed to deliver optimized mechanical properties, functionality, and printability with common FFF printers. The mechanism is printed from a thermoplastic polyurethane; (TPU) filament which was selected for its flexibility, which is necessary for the proper functioning of the spring elements. To characterize the designed mechanism, a tensile test is carried out to assess the holding force of the ILM. The force-displacement profiles are analyzed and categorized into distinct phases, highlighting the interplay between spring deformation, sliding, and disengagement. Finally, from the measurements of multiple printed specimens, a representative holding force is determined through averaging and assigned to the mechanism. The resulting tolerance, which can be attributed to geometric and material-related factors, is discussed. The testing results are discussed and compared with a numerical simulation carried out with a frictionless approach with a nonlinear Neo-Hookean material law. The study underscores the importance of meticulous parameter control in three-dimensional (3D) printing for the consistent and reliable performance of interlocking metasurface mechanisms. The investigation leads to a scalable model of an ILM element pair with distinct three-phase snapping characteristics ensuring reliable holding capabilities.

**Keywords:** additive manufacturing; FFF printing; printing parameters; experiment; interlocking metasurfaces; manufacturing-driven design



## 1. Introduction

Open source in the field of extrusion-based additive manufacturing; (AM) offers possibilities for customization, innovation, and economic manufacturing by manufacturers [1–3] and producing consumers (prosumers) [4]. The principles of open source can be traced back to Bruce Perens [5] and his work *The Definition of Open Source* from 1999. Back in the days of this work, the definition solely addressed computer software. However, nowadays, the term is much broader and includes additional areas such as open-source hardware [6–8], open innovation [8,9], and the increasing use of CAD models in the three-dimensional; (3D) printing community [10,11]. One of the major advantages of the open-source 3D printing community is an economic way of producing individual and customized products, which can lead to the sustainable development of ideas, projects, and, therefore, manufacturable items [4,12].

Extrusion-based AM is one of the most used AM methods and is commonly referred to as 3D printing [10,13–15]. Complex designs, which can be challenging or even impossible to manufacture using conventional methods, can be fabricated using extrusion-based

approaches, e.g., FFF [16–19]. However, owing to the layer-wise deposition of materials, it also requires certain design constraints to be printable, durable, and, foremost, functional [15,18,20]. This might be challenging for individual designs with smaller or more fragile components that require high printing resolution and accuracy [19,21] and where post processing, such as sanding and removing imperfections of the printed part, may be impeded. Thus, a manufacturing process-driven design appears to be essential for efficient and reliable production using material extrusion manufacturing such as FFF [22,23].

ILMs, which are a type of architectured locking surfaces, can serve as an example of this shape complexity. Here, naturally, non-occurring structured surface pairs are designed that can be temporarily connected or permanently joined to open up new mounting possibilities between surfaces. These pairs can be constructed with similarly shaped (androgynous) or with topologically different features (e.g., tongue and groove) [24]. Properties of parts manufactured using FFF may differ from objects fabricated through alternative processes, such as injection molding [18,19,25].

Accordingly, there is also an increasing academic interest in investigating the basic properties of 3D-printed parts to analyze appropriate handling of the limitations and errors of different extrusion-based AM machines [26].

ILMs have already been investigated by several research groups. For instance, Young et al. [27] conducted experiments and simulations on T-slot sliding mechanisms and their implementation in aerospace technology. Additionally, Peralta Marino et al. [28] published a study on modeling and testing interlocking structures fabricated with additive layer manufacturing processes (ALM). Topology-optimized ILMs with sliding characteristics have also been investigated by Brown et al. [29].

In this work, a similar approach to the aforementioned studies was adapted, and a printable ILM was designed and investigated. In contrast to Young et al. [27] and Peralta Marino et al. [28], the mechanism investigated in this study does not exhibit sliding properties; instead, it utilizes a snapping mechanism, where significant deformation occurs during assembly and disassembly, and only minor deformation occurs during loading when locked together.

These mechanisms can be identified by their distinctive clicking behavior. Consequently, the locking characteristics significantly differ from those of sliding mechanisms, where deformation is not the driving force behind interlocking; rather, it results solely from the rigid geometry. The snapping interlocking mechanism developed within the scope of this work, named *ShroomLock*, consists of pin and spring elements that lock together. The pin elements securely snap into the spring elements, establishing a reversible connection between the two components. Once locked together, these elements can withstand forces up to a specific threshold, beyond which the unsnapping phase initiates. This capability enables the temporary fastening of two bodies to each other. To establish a connection between these two bodies, both elements have to be distributed over the contacting surfaces, thereby creating two corresponding metasurfaces. Due to the element-wise nature of this ILM, the mechanical features of the surface connection can be traced back to a single representative cell. This drastically simplifies the study of this ILM.

This manuscript explains and evaluates the design of the newly developed ILM *ShroomLock*. For this, a testable representative cell is 3D-printed using FFF as the manufacturing method. Experimental and numerical testing is carried out to refine the mechanism's geometry. To achieve an elastic snapping connection, a flexible filament material is used for fabrication. TPU is characterized by its higher elongation at break and elasticity compared with polylactide; (PLA)-based filaments. This behavior can be advantageous for the necessary deformation during assembly of the mechanism [30].

The manufactured parts are tested for their holding force in the closing direction in tensile tests on a Zwick/Roell zwickiLine z2.5 kN testing machine (ZwickRoell GmbH & Co. KG, Ulm, Germany). This allows for the characterization of a force-displacement curve, and the possible maximum holding force of a representative element of the mechanism

can be determined. Numerical simulations, performed using the open-source library FEniCSx [31,32], serve as a foundation for optimizing the initial design and for the further development of this or similar locking mechanisms. All measured data, design files, and the code for numerical testing are available in a public repository which is linked in the data availability statement.

## 2. Materials and Methods

### 2.1. Additive Manufacturing-Method-Driven Design

To obtain the optimal shape of the pin and spring elements (see Figure 1a), the possibilities of the printing methods have to be taken into account. The design goal is to achieve a high strength, i.e., large holding force, and a reliable snapping behavior. However, the shape that allows for great snapping performance and provides the best holding power may not be printable at all. The design of the interlocking metasurface is therefore mostly driven by the choice of the manufacturing process, and the elements have to be optimized for strength and snapping under the restrictions given by the manufacturing method. As the intended mechanism is aimed to be an open-source project and printable for the majority of users, e.g., in the 3D printing community, the FFF method is used since it is one of the most commonly used 3D printing approaches.

The ILM elements are designed to be reasonably small, i.e., height and width are smaller than 10 mm, to maximize adaptability to specific applications. Here, the strength of the pin element has to be considered with regard to its size. A smaller pin will result in a smaller holding force. However, a smaller pin will also allow for a higher distribution density on the targeted surface, which results in a higher holding force over the surface. Smaller pins will also distribute the loading more evenly over the contacting surfaces. The choice to design the smallest possible pin was therefore made, where the minimal pin size was restricted by the available printer's characteristics. Aspects such as the line width when printing had to be considered for the minimum pin diameter; these key values which the pin dimensions are tied to are listed in Table 1.

**Table 1.** Technical requirements of the ILM mechanism.

| Category | Requirement |
|---|---|
| Connection properties | |
| | • Detachable connection |
| | • Restriction of all degrees of freedom between the surfaces |
| | • Snapping mechanism (deformation until form lock) |
| | • Force transmission based on form lock |
| Snapping conditions | |
| | • $r_2 > r_{\text{lamella}}$ |
| | • $h_1 \approx h_{\text{lamella}}$ |
| | • Spring elements made of flexible material |
| Design guidelines | |
| | • Overhangs up to 50° |
| | • Consideration of anisotropic material properties (build orientation) |
| | • Minimum diameter of vertical pins greater than 8 times the line width |
| | • Chamfer on the bottom edge to compensate over-extrusion in the initial layer |
| Production process | Fused filament fabrication (FFF), 0.4 mm Nozzle |
| Material | Thermoplastic polyurethane (TPU) |

The ILM was traced back to a representative individual mechanism cell for the design process and further studies. The individual mechanism comprises two components. On one side, there are radially arranged spring elements with an undercut feature between each lamella (see Figure 1a,b). These spring elements can be printed flat (horizontally), ensuring higher strength and consistency among the lamellae since the printing path of each element

will be similar, and fragile parallel filament tracks are avoided. For the functionality of the spring elements, elastic material properties as found with TPU are mandatory. On the other side, there is a pin specifically designed to interact with the spring elements. When the pin is inserted, it pushes the spring elements aside and guides them to the locking position. To maintain the desired locking performance over multiple cycles of engagement and disengagement, a proper fit between the pin element and the upper and lower faces of the spring element is necessary. On the one hand, this is ensured by the relation between $h_1$ and $h_{\text{lamella}}$ from Table 1. Here, to compensate for the tolerances in 3D printing and circumvent play between the two elements in the snapped state, $h_1$ is chosen to be smaller than $h_{\text{lamella}}$ in the design files (see Figure 1a). The chosen characteristic dimensions are given in Table 2. After printing, this results in the snapping condition being fulfilled and both heights being roughly equal. This results from the fact that the first layers of the lamellae are also the first layers on the print bed, resulting in a thinner profile. To ensure the relation between both heights, proper bed alignment is inevitable. On the other hand, chamfers and tapers have to be added to the contacting edges to ensure a proper fit and prevent wear. A taper is added to the lower part of the pin element's neck to compensate for over-extrusion; this has to be matched with a chamfer on the lower face of the spring element to avoid contact between the two elements. On the upper part of the pin element's neck, a taper is added to ease the transition from neck to pin. No matching chamfer is added to the spring element to ensure a snug fit of the lamellae between the bottom surface contact and the upper surface's edge, resting on the upper pin taper. With this, over-extrusion in the initial layer can be compensated for and an undercut on the pin side can be avoided. The pin side of the representative mechanism is also printed parallel to its connecting surface. Although this may not be the ideal printing orientation, it is necessary to avoid support structures and, thus, enhance the printing quality. To strengthen layer adhesion, the pin itself is meant to be manufactured solidly. Preliminary tests at various pull speeds showed no visual impairments. A demonstrator was designed to show how the individual mechanisms could be arranged to work as an ILM, as shown in Figure 1c. For applying the *ShroomLock* geometry to more complicated objects, the printability has to be evaluated for each specific object shape.

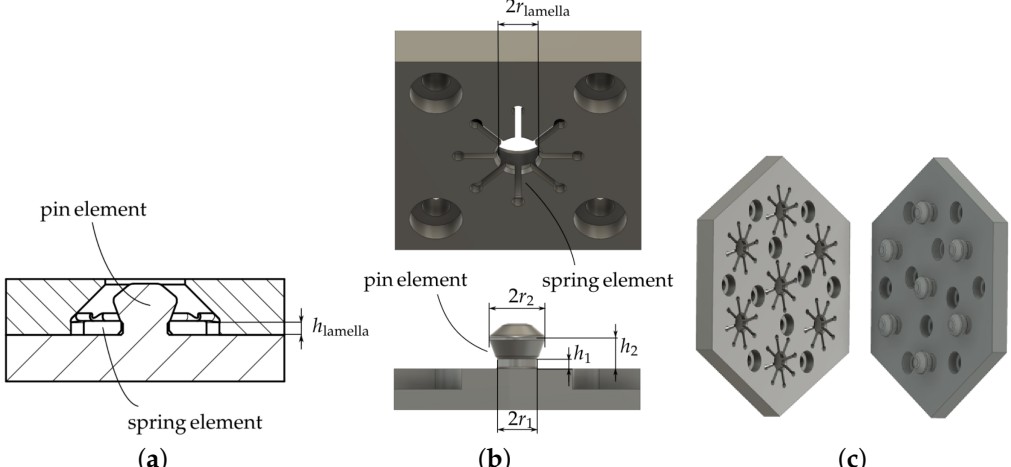

**Figure 1.** Design of the developed ILM: (**a**) sectional view of individual ILM cell in snapped state; (**b**) individual ILM mechanism cell in unsnapped state; (**c**) example of a hexagonal ILM in the unsnapped state. The locking mechanisms are distributed over the full contact surface area.

**Table 2.** Characteristic dimensions of developed ILM.

| Parameter | Value |
|:---:|:---:|
| $r_{\text{lamella}}$ | 2.6 mm |
| $h_{\text{lamella}}$ | 1.5 mm |
| $r_1$ | 2.4 mm |
| $r_2$ | 3.4 mm |
| $h_1$ | 1.2 mm |
| $h_2$ | 3.5 mm |

### 2.2. Additive Manufacturing Process

In initial approaches, the mechanism was printed with two different materials, PLA for the pin element and TPU for the spring element. The quality of the pins printed with PLA is significantly higher than that extruded with TPU; the latter requires the specific adjustment of print parameters and can be susceptible to imperfections and stringing in the extrusion printing process. However, one of the aims is to implement a workflow without changing filaments, and thus re-calibration of the printer and changing printer settings. Consequently, this leads to the choice of a flexible filament. Furthermore, the authors expected possible damage to the pin parts if printed with less flexible materials in case the matching surface elements are not aligned perfectly due to the user or printing discrepancies. Using TPU can allow for a greater tolerance regarding the range of the pin movement and thus decrease the possibility of damage to the parts. However, printing with TPU can cause issues during the creation of the print profile since flexible materials require a different approach than stiffer materials, i.e., retraction and flow rate, since the filament bends during retraction and compresses while being extruded outwards.

For the following investigation, a total of 8 pairs of the individual ILM cells were printed with an Anycubic Vyper FFF printer (Anycubic, Shenzhen, Guangdong, China). The FFF printer has a standard 0.4 mm nozzle. Due to its high elongation at break of 490% and relatively easy processability, the filament FLEX HARD (1.75 mm) from extrudr (Extrudr | FD3D GmbH, Lauterach, Austria) was chosen as the printing material. FLEX HARD (1.75 mm) is a hard TPU with a shore hardness of D58, which, to put the material into perspective, is harder than skateboard wheels. Ultimaker Cura (Ultimaker B.V., Geldermalsen, The Netherlands) was selected as the slicing software. For the manufacturing of the ILM mechanisms, an optimized TPU printing profile was created. The complete printing profile can be found in the provided repository (see the data availability statement below). The layer height is set to 0.1 mm, ensuring fine details and smoother surfaces. To prevent under-extrusion due to TPU's flexibility, the flow rate is increased to 103%, guaranteeing consistent material deposition. Enhancing strength and durability, the wall thickness is set to 2.4 mm, providing full material in the pins. To avoid potential issues with snapping, random z-seam alignment is employed, reducing the risk of consistent vertical z-seams. Aligning the top/bottom thickness with the height $h_{\text{lamella}}$ of the spring elements at 1.5 mm ensures maximum strength. To improve print quality, combing mode is activated, ensuring that the nozzle maintains contact with the printed object. To further prevent possible quality-reducing issues, the retraction settings and nozzle temperature must be adjusted properly. These measures have to be taken to optimize layer adhesion as well as to avoid imperfections such as blobs and stringing. The appropriate settings may vary between different printers and had to be determined experimentally for this study. A table of the key settings for the specific FFF machine used in this investigation is provided in the repository as well as in Table 3.

**Table 3.** Used printing parameters for printing the ILM on a Vyper 3D printer from Anycubic.

| Parameter | Value/Setting |
|---|---|
| Printer | Anycubic Vyper |
| Filament | extrudr FLEXHARD 1.75 mm |
| Layer Height | 0.1 mm |
| Wall Thickness | 2.4 mm |
| Top/Bottom Thickness | 1.5 mm |
| Z Seam Alignment | Random |
| Flow | 103% |
| Combing Mode | All |
| Print Sequence | One at a time |
| Initial Layer Speed | 15.0 mm/s |
| Infill Density | 20.0% |
| Infill Pattern | Gyroid |
| Printing Temperature | 215 °C |
| Temperature Initial Layer | 225 °C |
| Build Plate Temperature | 55 °C |
| Retraction Distance | 3 mm |
| Retraction Speed | 50 mm/s |

### 2.3. Experimental Tensile Test Series

The pairs of measurements, pin and spring element, are examined for their holding force in the direction of joining, i.e., orthogonal to the surface. The goal is to create a characteristic curve of the holding force over the displacement and determine a maximum holding force. To minimize potential changes in material properties resulting from storage and environmental factors, the individual components are 3D-printed from the same TPU roll, and the time interval between printing and testing is kept equally for both test sets. The experimental examination was conducted in two sessions, with four mechanisms tested during each session.

The tensile tests are performed on a Zwick/Roell testing machine (ZwickRoell GmbH & Co. KG, Ulm, Germany) of the zwickiLine z2.5 TN series with a linear measuring range of 2.5 kN. Further specifications are documented in Table 4.

**Table 4.** Machine parameters of the used Zwick/Roell testing machine.

| Parameter | Value/Unit |
|---|---|
| Linear measuring range | 2.5 kN |
| Sensor accuracy Class 0.5 | 0.5% |
| Sensor resolution (ADC) | 19 bit |
| Sampling rate | 100 Hz |
| Traverse speed | 2 mm/min |

Figure 2 depicts the experimental setup for the test series. To ensure the repeatability of the tensile test and minimize the impact of clamping forces on the mechanism, an insert was fabricated to serve as a carrier element for the spring and pin components. The insert was constructed from PLA due to its higher stiffness compared with the TPU used for the locking elements. This choice was made to reduce potential inaccuracies in the measurement of the holding force.

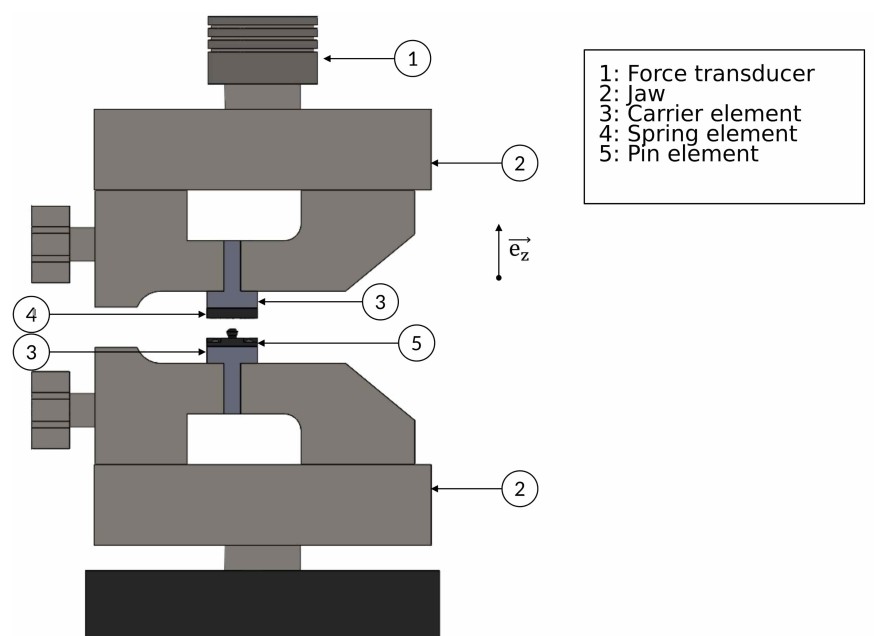

**Figure 2.** Setup for experimental testing of the ILM cell. Front view of the experimental setup with the locking elements loaded into the testing machine.

The test objects are screwed to the carriers and then connected for the first time using the snap mechanism. Initially, the assembled mechanism is clamped in the lower jaw, and the upper part of the testing machine is moved to the desired position, i.e., at the height of the carrier of the spring element. The system is tared before clamping the upper part. Eight mechanisms are investigated with four tensile tests each. Each tensile test consists of a full loading cycle up to the unsnapping of the pin from the spring element.

The pairs of elements are named consecutively with the printing material (TPU), the model version of the CAD model (VX), the pair of measurements (MY, 1 to 8), and the tensile test number (ZZ, 01 to 04). For example, the naming of the first pair of measurements of the seventh model version (V7)(M1) with the second tensile test (02) is shown:

TPU_V7_M1_02

*2.4. Numerical Modeling*

To assess the force and strain paths in the mechanism, as well as to help with future geometry optimization, a numerical analysis is carried out. The analysis may also be used to evaluate the mechanism's holding force. A corresponding boundary value problem (BVP) is formulated for the spring element ④. Its domain is discretized as $\Omega$. The BVP is expressed as a minimization problem, seeking the displacement field $\boldsymbol{u} : \Omega \to \mathbb{R}^3$ that minimizes the total potential energy $\Pi$. The spring element is modeled as a hyperelastic medium where the total potential energy is obtained as the sum of elastic stored energy $\Psi$ and the work of body and traction forces, $\boldsymbol{b}$ and $\boldsymbol{t}$, respectively

$$\Pi = \int_\Omega \Psi(\boldsymbol{u}) \, dV - \int_\Omega \boldsymbol{b} \cdot \boldsymbol{u} \, dV - \int_{\partial\Omega} \boldsymbol{t} \cdot \boldsymbol{u} \, dA \,. \tag{1}$$

The material is modeled with a nonlinear Neo-Hookean material law [33]

$$\Psi = \frac{\mu}{2} \left( \mathrm{tr}(\boldsymbol{C}) - 3 \right) - \mu \ln(J) + \frac{\lambda}{2} \ln(J)^2 \tag{2}$$

with the deformation gradient $F$, the right Cauchy–Green tensor $C$, the volume ratio $J$, and the Lamé parameters $\lambda$ and $\mu$, which, in terms of the Young's modulus $E$ and Poisson ratio $\nu$, are given as ([34], p. 186).

$$\lambda = \frac{E}{(1+\nu)(1-2\nu)}, \qquad \mu = \frac{E}{2(1+\nu)}.$$ (3)

From (1) the weak equilibrium formulation ($\delta\Pi = 0$) of the BVP is formulated. In the numerical analysis, a piecewise solution $u$ for varying $\delta u$ in a specific function space defined on the discretized domain is sought after. Two open-source software packages, GMSH [35] for meshing the domain and FEniCSx [31,32] for carrying out the numerical analysis, are used. Taking advantage of the symmetry of the spring element, the domain is reduced to a quarter of the actual component. This drastically reduces computation time. The corresponding mesh as well as the different boundary regions are depicted in Figure 3.

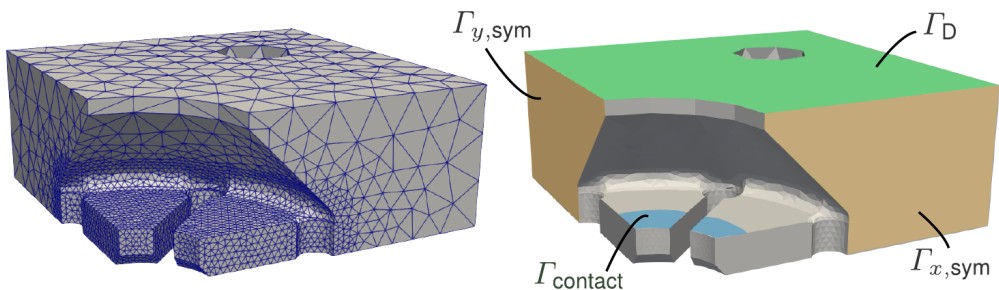

**Figure 3.** Discretized domain for the numerical analysis and designation of the boundaries. Symmetry conditions are applied on $\Gamma_{x,\text{sym}}$ and $\Gamma_{y,\text{sym}}$. The radius of $\Gamma_{\text{contact}}$ corresponds to the maximum radius of the carrier element $r_2$.

In the following, the BVP for modeling the unlocking of the mechanism is set up. The pin element ⑤ is assumed to be rigid and the contact during unlocking between it and the hyperelastic spring element is modeled. A frictionless approach is realized similarly as has been explained for the classical Hertzian contact in [36]. Speaking in analogy to the classical Hertzian problem, the pin element may be referred to as the indenter, as it is pulled out of the spring element similar to an indentation where the indenter is pushed into an elastic medium. The surface is approximately parameterized in the $z$-direction by a linear function and the gap between it and the spring element in dependence on the prescribed distance during unlocking is denoted as

$$h(r) = -u_{\text{pull}} + \frac{h_2 - h_1}{r_2 - r_1}(r - r_1), \qquad \text{for } r \leq r_2.$$ (4)

The parameterization allows for the formulation of the actual gap between the two components taking into account the $z$-displacement of the hyperelastic spring element

$$g(r) = h(r) - u_z(r).$$ (5)

In the contact area $\Gamma_{\text{contact}}$, the Signorini (see [36]) condition is formulated

$$\begin{cases} p = 0, & g > 0 \\ p > 0, & g = 0 \end{cases} \qquad \Rightarrow \qquad gp = 0.$$ (6)

If there is no contact between the components the gap is nonzero, and since the contact area is not loaded the pressure must be zero. If, on the other hand, the gap is zero, the contact area is loaded under the contact pressure, which is, in turn, nonzero. This condition

can be worked into the weak form by employing a large penalty parameter $k_{pen}$ and the Mackauley bracket $\langle\cdot\rangle_+$ giving the positive part of the gap

$$\int_\Omega \frac{\partial\Psi(\boldsymbol{u})}{\partial\boldsymbol{F}} \cdots \mathrm{grad}(\delta\boldsymbol{u})\,\mathrm{d}V = \int_\Omega \boldsymbol{b}\cdot\delta\boldsymbol{u}\,\mathrm{d}V + \int_{\partial\Omega}\boldsymbol{t}\cdot\delta\boldsymbol{u}\,\mathrm{d}A + k_{pen}\int_{\Gamma_{contact}}\langle u_z - h\rangle_+\,\mathrm{d}A\,. \quad (7)$$

Due to the nonlinearity of the problem in the contact formulation as well as in the Neo-Hookean material law, the weak form (7) is solved incrementally, and $u_{pull}$ is updated in small steps. The full BVP, including the realization of the symmetry conditions, reads as

$$\begin{cases} \boldsymbol{u} = \boldsymbol{0}\,, & \text{in } \Gamma_D \\ \boldsymbol{u}\cdot\boldsymbol{e}_x = 0\,,\ \boldsymbol{t}\cdot\boldsymbol{e}_y = 0\,,\ \boldsymbol{t}\cdot\boldsymbol{e}_z = 0\,, & \text{in } \Gamma_{x,sym} \\ \boldsymbol{u}\cdot\boldsymbol{e}_y = 0\,,\ \boldsymbol{t}\cdot\boldsymbol{e}_x = 0\,,\ \boldsymbol{t}\cdot\boldsymbol{e}_z = 0\,, & \text{in } \Gamma_{y,sym} \\ k_{pen}\langle u_z - h(u_{pull})\rangle = 0\,, & \text{in } \Gamma_{contact} \\ \boldsymbol{t} = \boldsymbol{0}\,, & \text{in } \partial\Omega \setminus \{\Gamma_D\,, \Gamma_{contact}\,, \Gamma_{x,sym}\,, \Gamma_{y,sym}\} \end{cases} \quad . \quad (8)$$

The penalty parameter $k_{pen}$ and the $z$-displacement $u_{pull}$ are chosen, the elasticity constants are provided by the filament manufacturer as

$$k_{pen} = 1\times 10^{14}\,,\ u_{pull} = 4\,\mathrm{mm}\,,\ E = 40\,\mathrm{MPa}\,,\ \nu = 0.45\,, \quad (9)$$

and $u_{pull}$ is updated in 200 time steps.

## 3. Results

The focus of the evaluation will be first on the experimental results of a case study of one specimen of the printed mechanism pairs. This includes presenting all four measurements of that specimen along with the mean calculation and the respective 95% confidence interval between different measurements of the unsnapping. Then, the results for multiple different specimens are compared with each other to evaluate and discuss the repeatability of the production of the ILM mechanism. Finally, an average of all individual means over the different printed mechanisms along with the confidence interval is calculated, and a correlation between the force profile and geometry is established.

### 3.1. Examining the Experimental Results

For each mechanism, a diagram with the force-displacement curves from the four tests is created, as shown in Figure 4a, representatively for the third test pair. What is immediately noticeable is that the first measurement—and with that the very first snap the mechanisms endured—has a significantly higher unlocking force. The same behavior is also shown by all other printed pairs. This systematic behavior suggests that after the first load cycle, which consists of the first snap to bring the elements together and then the first unsnap to separate them again, the elastic shakedown is reached [37]. For all further loading cycles, no continued accumulation of plastic strain occurs, as the unchanging force curves for the remaining measurements suggest. The material undergoes at first a slight plastic deformation where a higher force is needed, but then settles in an elastic response after the shakedown has been reached where a lower force is needed to reach the same displacement. To examine the mechanical features of the locking mechanism in its working range, only the measured data for the loading cycles after the elastic shakedown has been reached are considered. Measurements preceding the elastic shakedown were excluded from all further analyses.

The three remaining curves of each test pair are averaged (full-colored curve) and presented including the standard deviation within a 95% confidence interval (shaded areas), as illustrated in Figure 4b, representatively for the third mechanism. In this case, a nearly imperceptibly low standard deviation is evident in the range of 0 mm to 5 mm. Similar behavior with relatively small standard deviations can be observed across all test

series. For comparison, the individual means and SDs of the respective test mechanisms are displayed in one diagram in Figure 5a. It is apparent here that the measurements of TPU_V7_M2, especially between 4 and 5 mm, significantly stand out compared with the other mechanisms. Possible explanations include microscopic deviations in the shape of the elements due to their different locations on the printing bed during manufacturing. Additionally, these differences may lead to slight positioning variations of the mechanism on the testing machine. This, in turn, can result in the lamella element not being perfectly aligned with the pin element during the tensile tests, leading to a higher maximum holding force when pulled apart.

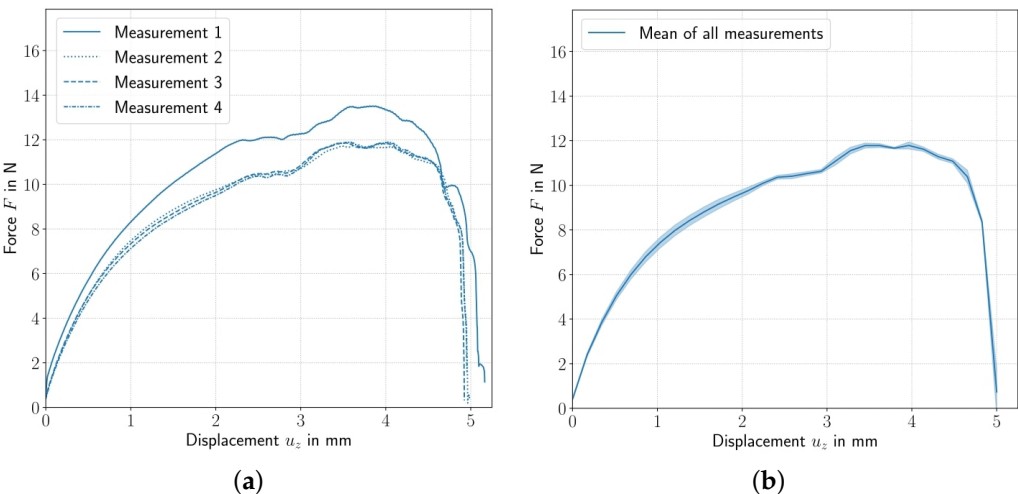

**Figure 4.** Results from the tensile test of the locking mechanism; only the specimen TPU_V7_M3 is considered. (**a**) Experimental results of the testing of part 3. The curve describes the relation between the pulling force and the corresponding shift of the pin element during unsnapping. The part underwent no snapping before the testing, and the elastic shakedown is visible for the first measurement. (**b**) Measurements 2–4 of part 3 averaged and plotted alongside the standard deviation.

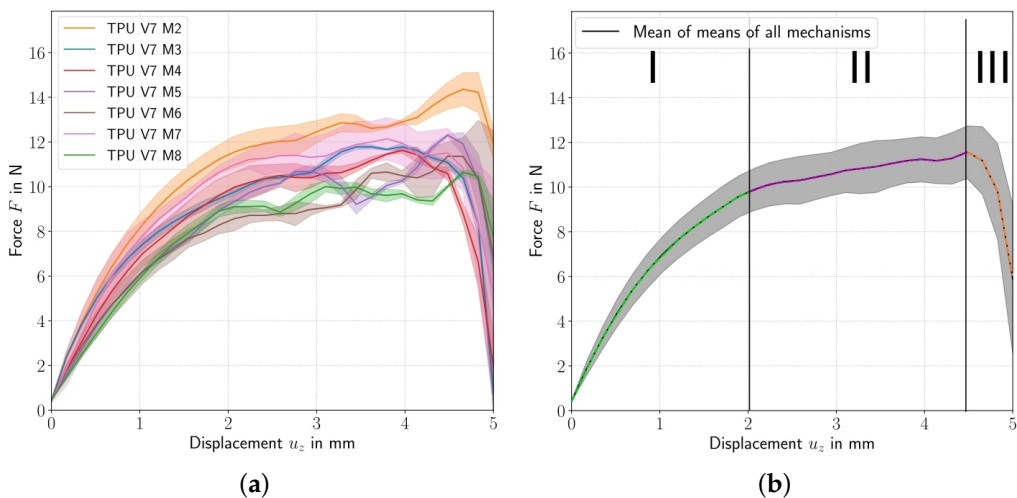

**Figure 5.** Results from the tensile test of the locking mechanism; different printed specimens are considered. (**a**) Averaged curves of mechanisms 2 to 8. Each mean curve is a result of the second up to the fourth snap of the respective mechanism. The solid line represents the mean force and the shaded area represents the respective confidence interval. (**b**) Average curve of the average curves of mechanisms 1 to 8. The division of the average curve into three regions (I, II, and III), as well as the coloring (green, pink, and orange, respectively), refers to the different snapping phases. The shaded area represents the confidence interval, taking into account the different measured specimens.

### 3.2. Geometry Force–Profile Relationship of the Locking Mechanism

Due to the variations among the averages of individual measurement pairs, an additional mean value is calculated from the individual means to determine the total holding force and its characteristic profile. This procedure results in a force-displacement curve as shown in Figure 5b. In this figure, the force profile is segmented into three distinct regions, enabling a correlation between the force profile and the geometry of the representative mechanisms.

In region I, the lamellae adhere to the bevel of the pin, causing a deformation that establishes a spring force in the *z*-direction (see Figure 6a). Here, no relative displacement between the pin and the spring element occurs.

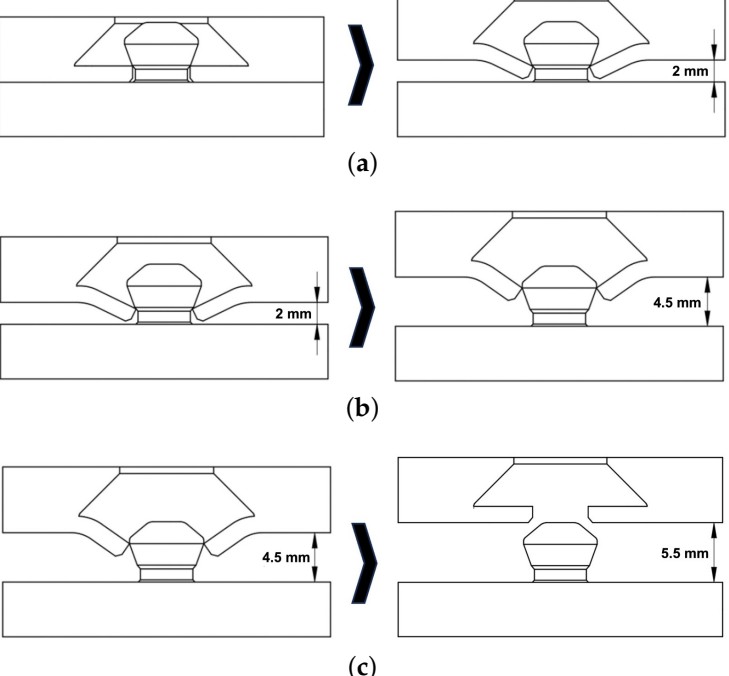

**Figure 6.** The unsnapping of the mechanism is visualized. The process can be split into three distinct phases. (**a**) Region I, the holding phase with stable equilibrium. (**b**) Region II, unsnapping begins. This state does not have a stable equilibrium. (**c**) Region III, the elements are separated.

In region II, the deformation of the lamellae increases further, such that the spring force is large enough to move the spring element over the pin element's bevel. The sliding of the lamellae along the pin results in a frictional force opposing the direction of motion, which decreases the curve's slope. Additionally, the lamellae continue to deform as the pin widens upwards, leading to a reaction force in the radial direction and a reaction force in the *z*-direction (see Figure 6b).

In region III, the mechanism disengages, the holding force reduces to zero, and the lamellae return to their initial position (see Figure 6c).

This distinct phase separation allows for a reliable characterization of the ILM. The holding force for the mechanisms is assigned to region I. Considering the standard deviation within the 95% confidence interval, an average maximum holding force for this region was measured of roughly $10\,\text{N} \pm 0.4\,\text{N}$ to $0.8\,\text{N}$. Because of a significant maximum deviation of roughly 8%, it is advisable to account for a safety margin. Consequently, a maximum overall holding force of $8.5\,\text{N}$ is determined for a representative element of the presented type of ILM. In region II, the mechanism starts to lock out, and in region III, the mechanism is completely locked out. It is noticeable that the general trend follows a smooth trajectory; however, the mean profile of individual mechanisms occasionally displays minor force fluctuations within region II. In Figure 7, it becomes apparent that the precision of the pin varies. Especially in the area where the lamellae slide over the pin and the force curve

settles into region II, some grooves are noticeable. This behavior is explained by the fact that individual lamellae jump into the grooves of the imperfections in the pin, leading to a brief relaxation and a slight reduction in the spring force in the *z*-direction.

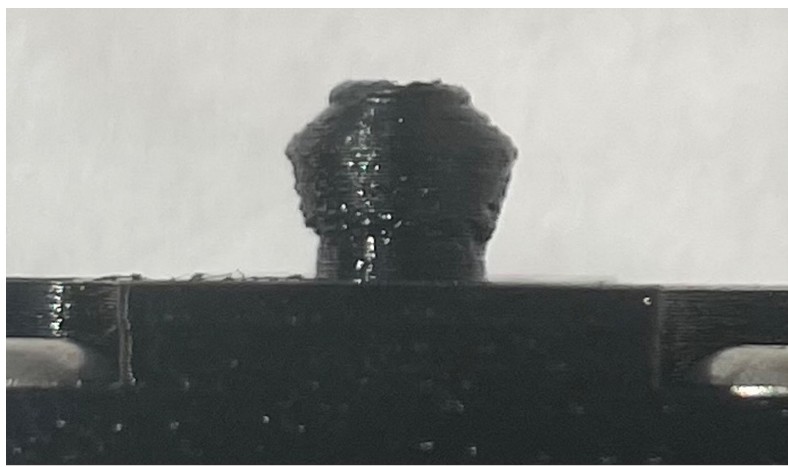

**Figure 7.** Picture of the pin element of TPU_V7_M8. Here, the surface roughness and printing accuracy, which is a result of the printing tolerances, can be seen.

### 3.3. Error Analysis for the Printed Specimens

As stated, there are some artifacts and imperfections in the appearance of the test objects, noticeably on the pin side. This could result from the choice of material, paired with the fabrication method, and lead to different behavior of the holding mechanisms between several prints. This is demonstrated by the partially erratic curve in region II of the individual test pieces. In any case, the FFF printing method—especially with printers that may not meet the manufacturing quality of an industrial-grade device—and filament material each can involve differences, e.g., between each print, material batch, or by changing environmental influences. Thus, material and printer influences might be investigated to narrow down the causes. While all mechanisms showed similar curve characteristics, variations in holding forces were measured. These differences can be attributed to discrepancies in alignment and centering during the tensile test series, as well as printing imperfections. In the investigation of the first four specimens (TPU_V7_M1-4_XX), the curves of mechanism TPU_V7_M1_XX exhibited significantly higher profiles compared with the results of the other specimens TPU_V7_M2-4_XX. This discrepancy can be attributed to misalignment; therefore, the measurements of TPU_V7_M1 were excluded from further study. Multiple tests of the same printed specimen demonstrate only minimal variations in the force profile. However, manufacturing tolerances result in distinct force curves for each individual printed specimen, as shown in Figure 5a. Characterizing the force profile and relevant holding force with individual measurements would only provide specimen-specific data. To offer a more comprehensive assessment of the mechanism's performance within a confidence interval, the mean values across various specimens are averaged. The deviation in individual means from the overall mean can reach up to 8%. While this approach may dampen fine details in the curve, it allows for the determination of characteristic parameters and the establishment of relationships between the force-displacement profile and the geometry of the mechanism itself. For comprehensive documentation, all data related to the tensile tests have been archived in the repository.

### 3.4. Examining the Numerical Results

Results of the numerical simulation are shown in Figure 8.

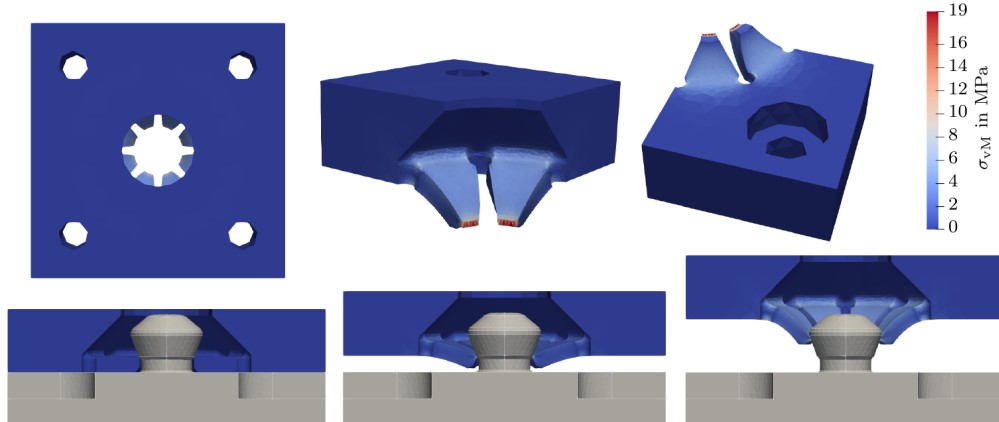

**Figure 8.** Results for the contact formulation. The full spring element can be assembled by exploiting the symmetry conditions and simply reflecting the obtained result. At the bottom, three time steps of the incremental load procedure are shown. For each, the von Mises stress is colored. The pin element is not part of the numerical study and is only visualized to give a reference.

Examining the resulting displacement field, a good fit is noticeable. However, the strong limitations of this approach also become clear. As the contact area $\Gamma_{\text{contact}}$ is assumed to be equal to the maximum radius of the pin element, highly concentrated stress appears in $\Gamma_{\text{contact}}$. Compared with the actual loading which the spring element endures, this is rather unrealistic and the stress should be concentrated over a larger area. This limitation may be overcome by incrementally changing the contact area as the load increases. In that case, the reference position should always be set to the last resulting deformed mesh. This would then affect the gap calculated between the pin and spring elements. Another aspect that should be mentioned is the choice of the penalty parameter. In this work, it is chosen as a constant; however, as it should directly correspond to the contact pressure, a variable formulation where the solution is not only sought for the varying field $\delta u$ but also a varying pressure field $\delta p$ is worth considering, see, e.g., [38].

The force necessary to unsnap the components is found to be roughly 22 N; this differs significantly from the force measured in the experimental section of this work. The discrepancy is, however, easily explained by considering the origin of the material parameters. Manufacturers of 3D printing filaments give out the elasticity parameters for a normed and injection molded part. These parameters help compare different filaments as they give relative information between two or more different filaments; if one material is stiffer than the other the provided elasticity parameters will certainly tell. The parameters, however, do not directly correspond to the actual parameters of the printed parts. On the one hand, the mismatch between the filament parameters and the parameters of the printed part is due to the difference in manufacturing. The normed, injection molded part undergoes different environmental influences compared with the 3D-printed part. On the other hand, and probably more important to explain the mismatch, the difference in density of the part resulting from the different manufacturing methods is substantial. See, e.g., [39] for a treatment of this in the context of PLA printing.

The same should also be considered when assessing the maximum von Mises stress found in the numerical analysis; observing Figure 8, stresses with up to 19 MPa are found. This value, however, should not be directly compared with the TPU's yield strength as the failing mode of the printed part should not directly correspond to the failure of the pure material but a delayering of the printed layers. The distribution of the stress in the part, however, still provides accurate information. Here, the numerical analysis helps to optimize shapes where large changes in stress occur which result in a higher wear in these areas. This may be used when working on further changes to the mechanism.

To give some concluding remarks on the numerical modeling, it should be mentioned that the modeling presented in this section does not claim to fully represent the actual

contact. Shortcomings of the used approach were discussed and suggestions for a more involved model were made. The code developed for this work is firstly intended to serve as the starting point for a rigorous modeling of the contact for ILMs. The simplified model did, however, result in good estimates in terms of the distribution of the maximum stresses. Regarding the estimation of the maximum load, for accurate estimation, effective material parameters are still required, as discussed above. The code for the further development of the contact modeling can be found in the GitHub repository linked in the data availability statement. Of special interest in further work is the adjustment of the contact pressure represented by the penalty parameter and a more suitable approximation of the contact surface.

## 4. Discussion

### 4.1. Holding Force of the Mechanism

In the experimental testing, the holding force of one pin and spring combination was investigated. As the intended use of this locking mechanism is to assemble an ILM, as in Figure 1c, an important characterization of the mechanisms is the holding force per area. For this, the characteristic dimensions of the locking elements are to be considered. Here, the spring mechanism is the limiting factor. A single spring takes up $2.5\,\mathrm{cm}^2$, meaning that the ILM has a holding force density of $4\,\mathrm{N/cm}^2$. The behavior of the mechanism when distributed over a surface is, however, deeply dependent on the type of loading, e.g., an unevenly distributed pressure will yield a different unlocking answer across the surface.

### 4.2. Prospective Applications

The potential applications of this mechanism are diverse, offering innovative solutions for a variety of settings. For instance, it can be configured to be used as a crucial component in modular design, e.g., as a versatile tool wall in a kitchen, an adaptable fixture in an automobile, or a practical element in space-related equipment. Furthermore, it can serve as a reliable wall or ceiling mount for various devices, such as video projectors. The adaptability of this mechanism opens doors to a wide range of future applications in different fields. Open-sourcing the design and associated data accelerates progress in the field of ILMs, enabling the 3D printing community to develop more specific applications of the mechanism.

### 4.3. Potential Improvements

The designed mechanism allows for potential improvements. To achieve a narrower confidence interval, one option is fabricating the pins from different materials to enhance the printing accuracy. Another possibility is adapting the mechanism to work with screws instead of printed pin elements, mitigating the surface effects caused by rough printing results, as discussed in Section 3. For further research, the developed mechanism offers many areas that warrant further exploration. On the one hand, there is the numerical modeling of the snapping mechanisms. An open-source-based FEM code with a contact formulation was established. The model shows a good approximation of the contact but relies on hard simplifications. Suggestions for a more rigorous model were made and further development of the code was motivated. The need for effective elastic parameters for the printed part also became clear. On the other hand, when it comes to testing the developed mechanisms, only the unlocking behavior was tested. Additional aspects that were not extensively tested, such as sustained loads on the closed mechanism or repeated snapping and unsnapping cycles, which are critical for ensuring the mechanism's long-term performance, are yet to be tested. Expanding the tests and investigating how the mechanism behaves after a certain number of interactions and identifying the number of interactions that lead to any significant changes, as well as determining if the holding force diminishes over time in a sustained loading test, can be performed in future research. Furthermore, exploring the scalability of the mechanism is of great interest. Understanding how the mechanism behaves if distributed over a surface would also offer valuable insight

for various applications. Here, the stiffness of the pin surface will greatly affect whether the holding force scales linearly as more pins are added to a surface.

## 5. Conclusions

In this work, the design and developing process of an interlocking metasurface mechanism, which was named *ShroomLock*, was presented. The primary objective was to develop a surface mechanism that exhibits significant deformation during assembly and disassembly, yet experiences only minor deformation when locked together, a behavior which was characterized as snapping. The concept was successfully realized in a printable mechanism made of TPU.

The developed mechanism boasts several outstanding features, including scalability over almost arbitrarily shaped surfaces, ease of use, and reliable snapping behavior. As the complete surface mechanism can be traced back to a single snapping element, it becomes easily testable. Through experimental testing, consistent force-displacement characteristics were obtained and three distinct displacement or force ranges were found. This three-phase snapping characteristic ensures the mechanism's ability to securely mount objects.

The manufacturing-driven design and testing procedure outlined in this paper holds significant promise for improving interlocking mechanisms in various applications. This approach offers valuable contributions to engineering practices and practical implementations. To help with future research, a GitHub repository was set up. It can be found in the data availability statement below. It includes the code for numerical testing, all measured data from the experiments, as well as the design and printing files.

**Author Contributions:** Conceptualization, L.N.S., P.G., H.L.F., P.W. and C.V.; methodology, L.N.S. and P.G.; software, L.N.S. and P.W.; validation, P.G. and L.N.S.; formal analysis, P.G. and L.N.S.; investigation, H.L.F., L.N.S., P.W. and P.G.; resources, C.V. and P.W.; data curation, L.N.S. and P.G.; writing—original draft preparation, L.N.S., P.G., H.L.F. and P.W.; writing—review and editing, P.G., L.N.S. and C.V.; visualization, L.N.S., P.G. and P.W.; supervision, C.V.; project administration, C.V.; funding acquisition, C.V. All authors have read and agreed to the published version of the manuscript.

**Funding:** We acknowledge support by the German Research Foundation and the Open Access Publication Fund of TU Berlin.

**Data Availability Statement:** All measured data, design files for printing the locking mechanisms, and codes for numerical testing are available in the linked GitHub repository https://github.com/lschek/ShroomLock.git (accessed on 24 October 2023).

**Acknowledgments:** The authors would like to thank Arion Juritza (TU Berlin) for assisting in conducting the experimental test series.

**Conflicts of Interest:** The authors declare no conflict of interest.

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
