# Peer review of "Extrusion-Based Additive Manufacturing-Driven Design and Testing of the Snapping Interlocking Metasurface Mechanism ShroomLock"

_inventions, doi:10.3390/inventions8060137_

Round 1

Reviewer 1 Report

Comments and Suggestions for Authors

The manuscripts describes about 3D printing driven design. Below are my comments:

Mention some mathematical error calculation values in abstract section.

Introduction needs revision. Follow the methods as found in latest 3D printed papers. 

Which FDM machine you have used. Mention the picture of machine and filaments description as applicable. 

How accurate is the mean of all measurement?? Can't we just take single measurement values??

Fig 4(a) and (b) need revision. The excess margin is confusing. 

Error analysis in percentage values!!!

Provide the comparison table as needful

Mention the conclusion

Comments on the Quality of English Language

Its fine. 

Reviewer 2 Report

Comments and Suggestions for Authors

        This study presents the manufacturing process driven development of an interlocking metasurface mechanism (ILM) for Fused Filament Fabrication (FFF) with a focus on open-source accessibility. The presented ILM is designed to enable strong contact between to planar surfaces. The mechanism consists of spring elements and locking pins which snap together when forced into contact. The mechanism is designed to optimize mechanical properties, functionality and printability with common FFF printers. The shortcomings of this paper are that there are problems in the naming and typesetting of many pictures in this paper. Secondly, the experimental process of 3D printing is not explained in detail, such as the key experimental parameters such as temperature are not reflected.

 1.The name of each picture in the article is missing, please check again, in addition, some pictures are not named, such as figure 6,8, please add.

2. Here the strength of the pin element has to be considered w.r.t. its size  Please prove in the experiment that the mass of the interlocking structure mentioned many times in the paper has a great correlation with the size. Please give an example of the ruler.

3. The number in Figure 2 is too large, and the size in the article is not appropriate, it is too obtrusive, please modify it.

Reviewer 3 Report

Comments and Suggestions for Authors

In this Manuscript entitled “Extrusion-based additive manufacturing driven design and testing of the snapping interlocking metasurface mechanism ShroomLock”, the authors presented a manufacturing process driven development of an interlocking meta-surface mechanism (ILM) for fused filament fabrication (FFF) with a focus on open source accessibility. The authors used thermoplastic polyurethane (TPU) filament to print ILM and evaluated the tensile strength of the printed mechanism.

Overall, the research article has some shortcomings, which need to be addressed before possible publication.

Please find the attached annotated file to see my comments.

Inventions journal publishes high-quality research articles. Based on my comments, the recommendation is Major Revision.

Comments on the Quality of English Language

Extensive editing of English language required

Round 2

Reviewer 1 Report

Comments and Suggestions for Authors

Thank you for the modification as suggested. 

Comments on the Quality of English Language

Its fine

Reviewer 3 Report

Comments and Suggestions for Authors

The article is in acceptable form, now.

Comments on the Quality of English Language

Proof reading,